# Preventing fear return in humans: Music-based intervention during reactivation-extinction paradigm

Ankita Verma[1], Sharmili Mitra[1], Abdulrahman Khamaj[2], Vivek Kant[3], Manish Kumar Asthana[1,4]*

1 Department of Humanities & Social Sciences, Indian Institute of Technology Roorkee, Roorkee, Uttarakhand, India, 2 Department of Industrial Engineering, College of Engineering, Jazan University, Jazan, Saudi Arabia, 3 Department of Humanities & Social Sciences, Indian Institute of Technology Kanpur, Kanpur, Uttar Pradesh, India, 4 Department of Design, Indian Institute of Technology Roorkee, Roorkee, Uttarakhand, India

* asthanakm@gmail.com, m.asthana@hs.iitr.ac.in

**Data Availability Statement:** The data that support the findings of this study are available on request. The data may be requested from the corresponding author or the institute human ethics committee.

## Abstract

In several research studies, the reactivation extinction paradigm did not effectively prevent the return of fear if administered without any intervention technique. Therefore, in this study, the authors hypothesized that playing music (high valence, low arousal) during the reconsolidation window may be a viable intervention technique for eliminating fear-related responses. A three-day auditory differential fear conditioning paradigm was used to establish fear conditioning. Participants were randomly assigned into three groups, i.e., one control group, standard extinction (SE), and two experimental groups, reactivation extinction Group (RE) and music reactivation extinction (MRE), of twenty participants in each group. Day 1 included the habituation and fear acquisition phases; on Day 2 (after 24 hours), the intervention was conducted, and re-extinction took place on Day 3. Skin conductance responses were used as the primary outcome measure. Results indicated that the MRE group was more effective in reducing fear response than the RE and SE groups in the re-extinction phase. Furthermore, there was no significant difference observed between SE and RE groups. This is the first study known to demonstrate the effectiveness of music intervention in preventing the return of fear in a healthy individual. Therefore, it might also be employed as an intervention strategy (non-pharmacological approach) for military veterans, in emotion regulation, those diagnosed with post-traumatic stress disorder, and those suffering from specific phobias.

## Introduction

Fear-related memories have adaptive and maladaptive effects on our lives [1]. For example, prior memories of a potential environmental threat help us survive because they enable us to identify and eliminate stimuli that could quickly harm our lives [2]. However, memory related to highly charged negative emotional experiences, like the death of a near or dear one, experiencing natural disasters like tsunamis or earthquakes, memories of war victims, rape

The contact information (email id) for the institute human ethics committee is ihec@bt.iitr.ac.in. The data are not publicly available as it contains information that could compromise the privacy of research participants.

**Funding:** The corresponding author is supported by the F.I.G. grant (IITR/SRIC/2741). The funders had no role in the study design, data collection and analysis, decision to publish, or preparation of the manuscript.

**Competing interests:** The authors have declared that no competing interests exist.

victims, etc., have a long-lasting adverse effect on the human brain which may eventually lead to excessive anxiety and stress [3]. Unfortunately, the memory related to these emotional experiences is long-lasting and enduring because it activates our body's sympathetic nervous system [4]. As a result, hormones like cortisol, noradrenaline, and adrenaline are released into the bloodstream, strengthening memory storage by enhancing noradrenergic activity inside the amygdala [4]. Thus, the memories can become intrusive and disruptive in these cases, causing distress and impairment in daily functioning [5].

Fear memories could be experimentally studied through the traditional Pavlovian fear-conditioning paradigm in which a neutral stimulus [NS] and a conditioned stimulus [CS] is paired with an aversive stimulus unconditioned stimulus [UCS]. After a few CS-UCS pairings, the CS becomes efficient at eliciting a fear-based conditioned response [CR], and, as a result, a fear memory is formed. Through extinction training (i.e., repeated presentation of non-reinforced CS), the fear response to the reinforced CS can be reduced or eliminated, demonstrating that the CS no longer predicts a threat [6, 7]. However, extinction has an inherent limitation as it does not erase or modify the existing fear memory traces. Alternatively, it creates a new safety memory trace, temporarily inhibiting the original fear association [6, 8, 9]. Therefore, the extinguished fear responses have been found to return leading to relapse in three conditions-spontaneous recovery, reinstatement, and renewal [6]. Spontaneous recovery is known as the increase in the strength of the initial CS-US association due to the passage of time. Renewal occurs if the conditioned stimulus is presented in a context different from the one in which extinction occurred, as extinction is context-dependent [8] Reinstatement is observed if the unconditioned stimulus is presented unexpectedly in the presence or absence of the conditioned stimulus, indicating a return of fear due to the occurrence of an adverse event following extinction [10].

Hence, researchers have attempted to augment extinction using pharmacological and behavioral techniques to prevent the return of fear responses. One of the techniques of augmenting extinction involves the reconsolidation of fear memories. Memory reconsolidation theory posits that when the memory is reactivated, the memory trace becomes temporarily malleable, allowing it to be modified [11, 12]. Therefore, if the fear memory is reactivated with a reminder of the CS, it becomes labile and subject to disruption during the reconsolidation window, i.e., 10 minutes to 6 hours after memory reactivation [12–15].

The first study targeting fear memory reconsolidation among humans involved the administration of beta-adrenergic receptor antagonist propranolol before reactivation, which reduced fear responses. It prevented the return of fear after 24 hours [16]. It was observed that propranolol selectively interrupted the protein synthesis of the fear memory in the amygdala, which resulted in reduced fear responses. However, the hippocampus-dependent declarative memory of the CS-UCS contingency remained intact [17]. However, researchers found that the long-term administration of propranolol has specific side effects like breathing problems, changes in blood sugar, hallucination, slow heart rate, sudden weight gain, drug tolerance, etc. [18]. It can even interact with other medications, vitamins, or supplements. This can be harmful or have serious side effects [19, 20]. Hence, an effective drug-free paradigm was needed to treat pathological memories. Schiller and colleagues [21] proposed a drug-free paradigm using a reconsolidation mechanism update. The authors demonstrated that a single brief exposure of a non-reinforced CS+ (a colored blue/yellow square paired with mild electric shock) used for memory reactivation just before extinction training could prevent the return of fear. These findings have important implications in the treatment of anxiety disorder; therefore, it has been further replicated in many other studies [13, 22–26].

Nevertheless, some studies also failed to show the effect of memory reconsolidation [17, 27, 28]. The inconsistency in successfully replicating the memory reconsolidation reactivation

paradigm might be due to boundary conditions [29]. Boundary conditions refer to circumstances that can either exhilarate or oppress a process [29]. In the context of memory reconsolidation, boundary conditions refer to the factors that influence whether a particular memory will undergo reconsolidation and the extent to which it can be modified. Some of the factors are memory strength [30, 31], age of the memory [32, 33], duration of retrieval session, and too short reactivation sessions fail to induce reconsolidation [32], whereas long non-reinforced reactivations lead to extinction [34–36]. Also, it has been reported that reconsolidation is constrained when reactivation occurs in a distinct context [37, 38]. Hence, this study attempts to reduce the fear responses and prevent the return of fear by introducing a music stimulus of positive valence and low arousal along with extinction during the reconsolidation window. As the emotional fear-related memories (with negative valence and high arousal) are temporarily destabilized and susceptible to update and modification during the reconsolidation window [12], exposure to music (with high positive valence and low arousal) during this reconsolidation window may reduce the negative emotional valence of fear-related memories and prevent the return of fear in humans.

Research has shown that fear memories involve intricate neural networks, primarily centered around the amygdala, a key hub for processing emotional stimuli and generating fear responses. The amygdala's lateral nucleus (LA) is particularly crucial in associating the conditioned stimulus (CS) with the unconditioned stimulus (UCS), forming the basis of fear memory formation. Furthermore, the central nucleus of the amygdala (CeA) orchestrates the expression of fear responses through its connections with brainstem regions responsible for autonomic and behavioral responses to threats [39]. Consolidation of fear memories involves complex interactions between the amygdala and the hippocampus. The hippocampus encodes contextual information and contributes to the formation of episodic memories related to fear-inducing events [40]. The prefrontal cortex, including the medial prefrontal cortex (mPFC), regulates fear responses through inhibitory control over the amygdala [41]. This neural circuitry's dynamic interplay contributes to the fear of memory consolidation, retention, and subsequent retrieval.

Regarding modulation, memory reconsolidation theory suggests that reactivation of a fear memory engages the same neural circuits involved in its initial formation. During reconsolidation, synaptic plasticity mechanisms and protein synthesis contribute to the updating and modification of the memory trace [11, 42, 43]. This offers a potential avenue for interventions like the one proposed in this study, involving the exposure to music during the reconsolidation window.

Music was selected as an intervention due to its emotion-modulating properties and efficacy in reducing anxiety. According to Gretsegger et al. [44], positive valence music therapy can help children and adolescents who have experienced trauma to learn and manage their emotions, as well as reduce the symptoms of depression and anxiety. Furthermore, listening to music decreases anxiety and physiological reactions to stress in people with post-traumatic stress disorder (PTSD) [45]. Similarly, Mehrent and group [40] explored the potential benefits of listening to positive music for individuals who have experienced traumatic events. They found that the experimental group that was getting music therapy showed significant improvements in symptoms of depression and anxiety. However, the control group showed no significant changes in these symptoms. These studies demonstrate that music has the potential to alter emotions and physiological responses to traumatic life events.

Therefore, the current study aims to establish a memory reactivation-extinction paradigm in the Indian context. It also seeks to determine the effect of music in regulating fear memory played during the memory reconsolidation window to prevent the return of fear.

## Material and method

### Participants

A priori power analysis using G.Power [46] with an alpha value of 0.05, power of 0.80, and a medium effect size of 0.25 [47], the sample size required 3x3 repeated measures ANOVA was 36 [48, 49]. Therefore, sixty participants (Males = 32, *M* = 26.96, *SD* = 1.67, Females = 28, *M* = 25.90, *SD* = 1.75) were recruited from the Indian Institute of Technology Roorkee through purposive sampling techniques. This sampling method was chosen for convenience and to minimize the likelihood of participant attrition [50]. The study was further briefly described to the interested participants, i.e., it is a three consecutive day study and will include a collection of physiological data of skin conductance response. No monetary compensation was provided to the participants for participating in the study. Participants were eligible for inclusion only if they met the following criteria: a low score on the anxiety scale, medication free from the last 12 hrs., no uptake of nicotine and caffeine or exercised two hrs. before the experiment, no medical illness, especially (heart diseases, asthma, thyroid, diabetes), no psychiatric or neurological disorders. No students with psychology as a major subject were included as participants. The data collection was conducted from 20[th] October 2022 to 30[th] December 2022.

Participants were then randomly assigned into three groups of twenty each, i.e., SE (Standard Extinction), RE (Reactivation Extinction), and MRE (Music Reactivation Extinction), and after that, their demographic details and written informed consent were taken. After data analysis, four participants were excluded as they did not volunteer to participate in re-extinction training (Day 3 paradigm). Due to some technical problem, the data of two subjects were invalidated, and four were excluded as they did not show any conditioned response. Hence, the data from 50 participants were used for statistical analysis. Ethical approval was taken from the Institute Human Ethics Committee (IHEC No- IITR/IIC/22/16). Participants were informed that they might withdraw from the research at any time. They were also told that the data would be used solely for research purposes and that only the researchers could access it.

### Materials

#### Stimuli

**Stimuli.** *Conditioned stimuli (CSs).* Two gray geometrical shapes (a square and an equilateral triangle, dimensions: 12 cm x 12 cm) [51] served as either CS+ or CS- presented for 4 sec. These two images were counterbalanced between the groups. The CS+ was paired with the unconditioned (UCS) on a 100% reinforcement schedule, and CS- was never paired with the UCS. E-Prime software (3.0) (Psychology Software Tools, Inc., Pittsburgh, PA, USA) connected to HP ProDesk 600 G5PCI MT was used for a stimulus presentation.

*Unconditioned stimuli (UCS).* The woman's screams (code number 276) from the International Affective Database System (IADS) for 2 sec were used as the unconditioned stimulus [26]. It was presented through noise-canceling headphones, with the intensity set between 90–96 dB to correspond to the participant's comfort level. A sound card was also used to improve the sound quality.

*Music stimuli.* The music stimulus was chosen based on a pilot study conducted for selection of appropriate stimuli for the study [52]. Ten music stimuli were chosen based on three genres and were instrumental in nature. The first category included Western classical music, including the famous works of Mozart and Beethoven music, like Symphony No. 40 [53–55], Serenade No. 13 [56, 57], and Fur Elise [58–61]. The second category of stimulus consisted of Indian Classical Music, like Raga Anand Bharavi [62, 63], Raga Yaman [64, 65], and Raga Mohanam [66, 67]. The third category of music was natural sounds. The natural sound of

blossom (sound of a chirping bird) [68–70], healing flute [71, 72], River Sound [73–76], and Tibetan healing sound [77–80] were used. One hundred and two healthy participants (60 males and 42 females; mean age = 24.95 years, SD = 4.62) were instructed to listen to music stimuli for 60–90 sec and rate them on a 9-point Likert scale on valence and arousal dimension. After analysis, it was found that the Healing Flute was rated as the highest valence for both males M = 7.03 (1.78) and females M = 7.00 (2.07) and lowest arousal for males M = 3.55 (2.86), and females, M = 2.95 (2.29). Hence, it was chosen as the music stimulus for this study.

## SCR recording and analyses

Skin Conductance Response (SCR) measures the changes in the electrical conductance of the skin, which is indicative of the activity of the sympathetic nervous system. In this experiment, the data of SCR was recorded using the Biopac MP160 (Biopac Systems, Inc., California, USA) system, a 16-channel direct current amplifier system using Acknowledge software (Biopac Systems, Inc., California, USA) at the sampling rate of 1000 Hz. To record data, two self-adhesive isotonic gel electrodes were attached to the medial phalanges of the participant's non-dominant hand [81]. SCR data was filtered with a 1 Hz low pass filter and was further segmented into different phases (e.g., habituation, acquisition, extinction, and re-extinction). After that, each segment was baseline corrected 1000ms before the onset of the stimuli and characterized by taking the maximum of the SCR deflection in the 1–5 sec interval after stimuli onset, consistent with the previous study [26].

## Procedure

All participants completed the experiments in the same environment with mild light and a temperature of 26 degrees Celsius [82]. It was ensured that the Biopac device was started ahead of time to verify that it was stable and functioning. In this experiment, the BIOPAC device has been used to collect SCR data, which measures changes in sweat gland activity that indicate psychological or physiological arousal [83]. This physiological monitoring was performed throughout the session on the participant's arrival. They were asked to answer the State-Trait Anxiety Inventory [84] and the Positive Affect and Negative Affect Scale [85] demographic details, and written informed consent was also collected.

## Experimental procedure

The present study will be conducted for three consecutive days [21, 86, 87], 20 hours to 24 hours apart (Fig 1). Initially, participants were provided demographic details, informed consent, and declaration form (medical condition, willingness to follow the study protocol: and compliance to abstaining from stimulants and physical activity before testing). After completing the questionnaires, they were randomly assigned to one of three groups: standard extinction, reactivation-extinction, and music reactivation-extinction. Participants were connected to the SCR recording apparatus in all conditions. Further, they received the following instructions before the start of fear conditioning: "We are going to start. There will be two images presented. The aversive noise of women's screams will be paired with any one of the images. "Please keep your eyes on the screen at all times."

**Day 1: Fear acquisition.** The first session of Day 1 consisted of a habituation and acquisition period in which participants learned the association of conditioned (CS) and unconditioned stimuli (UCS). During these two phases, two geometrical shapes (a square and an equilateral triangle, all gray in color, 12 cm in width and 12 cm in height) were presented to the participants on a PC monitor at a 16-degree visual angle in a randomized order for 4s with an inter-stimulus interval of 10-12s and counterbalanced as CS+, and CS-, so that both the

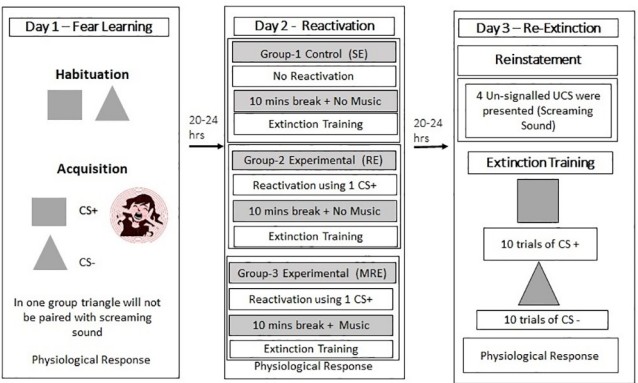

**Fig 1. Experimental design.** Fear conditioning on Day 1 was established by pairing a geometrical shape (square or triangle) with a screaming sound. On Day 2, a brief memory reminder CS+ was given to both experimental groups (RE and MRE). Following the reminder, the MRE group was subjected to a musical intervention after a 10-minute interval, and all three groups underwent extinction training. On the third day of the experiment, fear was reinstated through the implementation of four un-signaled unconditioned stimuli (UCSs) across all three groups, which was subsequently followed by extinction training.

square and equilateral triangle were equally often selected as CS+ and CS-. Stimuli were presented in a pseudo-randomized order (i.e., not more than two consecutive trials of the same CS+ in a row were repeated) using E-prime Version 3.

During the habituation phase, 8 trials of each geometrical shape (a square and an equilateral triangle) were presented. In the acquisition phase, 10 trials of each geometrical shape (a square and an equilateral triangle) were presented in pseudo-randomized order. Along with one of the two geometrical shapes, a woman's screams were presented co-terminating with the CS + for 2 sec [21] while the other geometrical shape served as CS-. The acquisition had 10 reinforced presentations of CS+ and 10 presentations of CS-. This treatment condition was given to all three group members.

**Day 2: Reactivation and music intervention.** On the second day, the first group of participants received no reminder trial, but extinction training followed by a 10-minute break [25–27]. The second group underwent reactivation, followed by extinction training (10 CS+, 10 CS-). In the reactivation phase, a single presentation of the CS+ for 4s without the UCS acts as a reminder trial. The third group received a reminder trial, followed by a 10-minute break, music stimuli, and extinction training (10 CS+, 10 CS-).

**Day 3: Re-extinction.** A day later (Day 3), 24 hrs later in the extinction training (i.e., day 2), the fear memory was reactivated by re-exposure to the four trails of the un-signaled unconditioned stimulus (screaming sound) for 4 seconds each to test the return of fear in humans. After that, all three groups underwent extinction training (10 CS+, 10 CS-) for a second time to trace the reinstatement of fear of previously learned fear memory contents. On all days, headphones and SCR electrodes were connected to all the participants, and the SCR was recorded throughout the experiment [26].

## Statistical analysis

The results were documented and analyzed using AcqKnowledge software, and the second software was SPSS version 21.0 (IBM SPSS Statistics, Chicago, IL, USA). Demographic data such as gender (male, female) was compared between Standard Extinction, Reactivation-Extinction, and Music Reactivation-Extinction Groups using a one-way analysis of variance

**Table 1. Demographic details of the participants of the three groups, (N = 50) SE (n = 17), RE (n = 17), and MRE (n = 16), along with the man Whitney U test to test the gender differences.**

| Groups | Sex | Age Mean (SD) | U | p |
|---|---|---|---|---|
| Standard Extinction (SE) [N = 17] | F (6) | 25.83 (1.72) | 18.0 | 0.10 |
|  | M (11) | 27.09 (1.64) |  |  |
| Reactivation-Extinction (RE) [N = 17] | F (8) | 26.37 (1.30) | 27.5 | 0.64 |
|  | M (9) | 26 (1.60) |  |  |
| Music Reactivation-Extinction (MRE) [N = 16] | F (9) | 26.12 (2.10) | 19.0 | 0.33 |
|  | M (7) | 27.28 (2.21) |  |  |

**Table 2. Psychometric data of the participants (with the standard deviation in brackets) for the three groups on the two questionnaires- Spielberger State-Trait Anxiety Inventory- Trait (STAI-T), Positive and Negative Affect Scale (PANAS).** PANAS$_1$ is a positive Affect, and PANAS$_2$ is negative Affect.

| Groups | SE | RE | MRE | F | p |
|---|---|---|---|---|---|
|  | M(SD) | M(SD) | M(SD) |  |  |
| STAI-T | 33.44 (5.42) | 33.87 (5.95) | 36.66 (8.03) | 1.14 | 0.32 |
| PANAS$_1$ | 23.16 (3.07) | 24.56 (2.33) | 24.73 (2.31) | 1.82 | .17 |
| PANAS$_2$ | 10.72 (2.02) | 10.5 (1.63) | 11 (1.46) | 0.31 | 0.72 |

(refer to Table 1), Psychometric data (PANAS and STAI) were analyzed using one-way ANOVA (Table 2).

The entire experiment was divided into five phases, i.e., habituation, acquisition, extinction, re-extinction, and reinstatement of fear. Differential SCR values (CS+ minus CS-) were calculated for each experimental phase (habituation, acquisition, extinction, and re-extinction) to evaluate fear conditioning [13]. A repeated measures ANOVA with phase (habituation, acquisition, and extinction) as the within-subject factor and intervention group (standard extinction, reactivation-extinction, and music reactivation-extinction) as the within-group factor was performed. To study the effect of reinstatement, we calculated the difference between the differential SCR values of the mean of the first two trials of the re-extinction phase (Day 3) and the mean of the last two trials of the extinction phase (Day 2) [26].

## Results

A repeated measures ANOVA with phase (habituation, acquisition, and Extinction) as the within-subject factor and intervention group (standard extinction, reactivation-extinction, and music reactivation-extinction) as the within-group factor indicated that there was a significant main effect of phase (habituation and acquisition) across all participants [F (1, 46) = 211.243, p<0.001, ηp2 = 0.821] (Fig 2). Post hoc Bonferroni's test for multiple comparisons revealed that there was a significant difference in the SCR responses in the habituation and acquisition phases [p< 0.001, 95% C.I. = (-0.107, -0.081)]. Further, we found that there were no significant group differences in the habituation and acquisition phases [F (2, 46) = 1.435, p = 0.249, ηp2 = 0.059]. This indicates that the participants in all the three groups had similar baseline fear responses during the habituation phase and had acquired equivalent levels of fear after the fear Acquisition phase.

There was a significant main effect of phase (acquisition, extinction) on the SCR responses [F (1, 46) = 100.085, p< 0.001, ηp2 = 0.685]. We found a significant interaction effect between

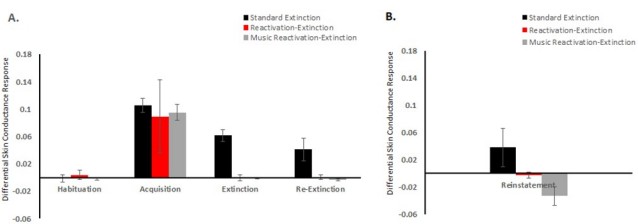

**Fig 2. Mean differential skin conductance response (SCR).** A. Mean Differential SCR for the Habituation Phase, Acquisition Phase, Extinction Phase, and Re-Extinction Phase. B. Mean differential SCR for Reinstatement (difference between the mean of the first two trials of the Re-extinction phase, i.e., Day 3, and last two trials of the Extinction phase, i.e., Day 2).

phase (acquisition, extinction) and group (standard extinction, reactivation-extinction, and music reactivation-extinction) [F (2, 46) = 4.653, p = 0.014, ηp2 = 0.168].

Our results revealed no significant difference in the SCR responses between the extinction and re-extinction phases [F (1, 46) = 0.659, p = 0.421, ηp2 = 0.014] and no interaction effect between phase (extinction and re-extinction) and group (standard extinction, reactivation-extinction, and music reactivation-extinction) [F (2, 46) = 1.848, p = 0.169, ηp2 = 0.074].

To further evaluate the effect of the intervention on the three groups (standard extinction, reactivation-extinction, and music reactivation-extinction), we conducted a one-way ANOVA with Fisher's LSD post hoc tests In the extinction condition, the difference in SCR between Group 1 (SE) and Group 2 (RE) was found to be statistically significant, p = .00, level. There was also a significant difference in SCR between Group 1 (SE) and Group 3 (MRE) at the p = .00 level. However, there was no significant difference in SCR between Group 3 (MRE) and Group 2 (RE) p = .95.

In the re-extinction condition, the difference in SCR between Group 1 (SE) and Group 2 (RE) was found to be statistically significant (p = 0.01). There was also a significant difference in SCR between Group 1 (SE) and Group 3 (MRE) at the p = .00 level. However, there was no significant difference in SCR between Group 3 (MRE) and Group 2 (RE).

In the reinstatement condition, we calculated the difference between the differential SCR values of the mean of the first two trials of the re-extinction phase (Day 3) and the mean of the last two trials of the extinction phase (Day 2) to study the effect of reinstatement. The results indicated that the difference in SCR between Group 1 (SE) and Group 2 (RE) was not statistically significant. However, there was a significant difference in SCR between Group 1 (SE) and Group 3 (MRE) at p = .01 level. There was also no significant difference in SCR between Group 3 (MRE) and Group 2 (RE).

## Discussion

The current study attempted to explore the efficacy of a music intervention in updating fear memories. In this study, we used a three-day differential fear conditioning paradigm to compare the effects of music reactivation-extinction, reactivation-extinction, and standard extinction on conditioned fear responses measured by SCR.

Our findings suggest a novel drug-free intervention for updating fear memories by using music of positive valence and low arousal within the reconsolidation window. Through our findings, we demonstrate that during the extinction and re-extinction phases, both RE and MRE group participants were more effective in attenuating fear-related responses than standard extinction. Although our results do not show a significant difference between the SE and

RE groups in the return of fear, we have observed a trend that indicates that the RE group may have been more effective in attenuating the return of fear responses compared to the SE group.

The difference in the effectiveness of extinction and reactivation-extinction in attenuating fear responses may be understood by the difference in the mechanisms between the two phenomena. Extinction aims at weakening the fear memory by generating inhibitory circuits (CS-no UCS) that suppress the fear responses [88]. On the other hand, memory reconsolidation updates the existing fear memory trace through reactivation followed by extinction. Our results demonstrate the failure in updating the memory using extinction only during the reconsolidation window. We may have failed to update the memory using reactivation and extinction only (RE group) due to boundary conditions of memory reconsolidation, like memory strength. Since we used a strong fear acquisition protocol with a 100% reinforcement level, the strong fear memory may have become resistant to disruption upon reactivation-extinction only [33].

As previously noted, certain studies using the reactivation-extinction paradigm, such as those conducted by Golkar and colleagues [27], Kindt and Soeter [17], and Wood and colleagues [28], have yielded inconclusive results regarding the effect of memory reconsolidation. Therefore, this study tried to provide a novel, non-invasive, drug-free behavioral safety learning technique to disrupt fear-related memories. In this new paradigm, fear memory is reactivated with one CS+ followed by exposure to music with positive valence and low arousal within the reconsolidation window. The findings suggest that during the phase of reinstatement, this music intervention effectively prevented the return of fear in participants in the MRE group.

The study was structured into five phases: habituation, acquisition, extinction, reinstatement, and re-extinction of fear. The results indicate that the mean differential SCR for the habituation and acquisition phases were statistically insignificant for all three groups (SE, RE, and MRE). Participants' responses to these treatment conditions were comparable because there was no difference in how they perceived or were conditioned by the stimulus [26].

The participants belonging to the RE (Reactivation Extinction) and MRE (Music Reactivation Extinction) groups exhibited a reduction in differential SCRs during the extinction and re-extinction phases. This finding lends support to the efficacy of the reactivation-extinction paradigm in preventing the return of fear in both conditions. However, in the case of reinstatement of fear, the combination of a reminder (CS+) and music during the reconsolidation window MRE group appears to be more effective in preventing fear-related responses. This study's findings indicate that music played during the reactivation window may be more effective in reducing human fear-related reactions. In the MRE group, playing positive valence and low arousal music, followed by extinction after reactivation, may have created specifically two distinct memories, i.e., a music memory and an extinction memory. Since the two memory traces were mood-congruent, it might have resulted in a unified encoding of music and extinction-memory traces. This unified encoding of the two memories may have created a more robust effect on the modification of the reactivated fear memory compared to reactivation and extinction alone [89]. Upon encountering the CSs after reinstatement of fear, the unified memory of music and extinction may have been activated, resulting in reduced fear responses. Therefore, through our study, we observed that a combination of music along with extinction was more effective in reducing fear responses, as well as the return of fear upon reinstatement on Day 3.

Previous studies have demonstrated that music has arousal-modulating capacities and may be used as a potential non-pharmacological treatment alternative to propranolol in reducing fear responses. Relaxing music has been reported to reduce anxiety and stress responses by decreasing heart rate and skin conductance levels. Music may also have the potential to counter emotional arousal [90–92]. In line with previous research, our study demonstrates the

efficacy of the combined effect of music and extinction memory reconsolidation over only reactivation-extinction. Also, our results could be attributed to the findings in music therapy literature, which propose that the affective qualities of music, pertaining to its emotional characteristics, can substantially impact an individual's mood and emotions. According to [36], the act of evoking an emotional response, whether positive or negative, is also possible among the listeners.

Moreover, the literature also suggests that some music genre has been found to be an effective intervention for relieving stress symptoms among asthma and cancer patients [93, 94] and reducing preoperative anxiety [95]. It has been found to aid in the treatment of psychiatric conditions like schizophrenia, depression, sleep disorders, and dementia [96]. As an adjunct to pharmacotherapy, it has effectively reduced anxiety and depressive symptoms among individuals with generalized anxiety disorders [97]. Consequently, in this experimental condition, music of positive valence and low arousal played during the memory reconsolidation interval may have affected and reduced the negative valence of fear-related memories [36]. Hence, our study significantly contributes to the music therapy literature and highlights the importance of using music as an intervention technique along with the reactivation extinction paradigm.

Further, our findings may be explained by the involvement of the brain regions in the processes of extinction, memory reconsolidation, and music processing. fMRI studies have provided evidence for the involvement of distinct brain regions in the processes of standard extinction and memory reconsolidation. The amygdala, anterior cingulate cortex (ACC), and ventromedial prefrontal cortex have been implicated in extinction [98], whereas memory reconsolidation involves the activity of the amygdala, hippocampus, and vmPFC [99]. In addition, the affective and cognitive processing of music is associated with the right posterior temporal lobe and the insula [100]. Therefore, the difference in the effect of standard extinction, reactivation-extinction, and reactivation-extinction, along with music, on the attenuation of fear responses observed in the current study may be traced back to the distinct interplay of brain regions in the three processes. The effect of music and reactivation-extinction may have been more effective in reducing fear responses due to the alterations in the limbic circuit.

Literature suggests that a combination of interventions post-reactivation is successful in attenuating fear responses. In detail, orally administering beta-adrenergic receptor antagonist propranolol within the reconsolidation window, followed by extinction, attenuated fear responses and prevented the return of fear after 24 hours [16]. Similarly, engaging in a working memory task with a high cognitive load during the reconsolidation window disrupts the fear memory and results in reduced fear responses and return of fear upon reinstatement [101, 102] administered r-TMS over the dorsolateral prefrontal cortex post-reactivation and observed reduced physiological fear responses and lowered return of fear upon reinstatement. Researchers have also used cathodal transcranial direct current stimulation of the right dorsolateral prefrontal cortex after reactivating the fear memory. However, the results demonstrated that cathodal tDCS did not successfully inhibit fear memories [103]. Hence, in the current study, we used music of positive valence and low arousal during the reconsolidation window due to its arousal-modulating properties. Our results indicate that reactivation-extinction, along with music, was more effective in reducing fear responses, as indicated by SCR arousal.

Although the current study contributes to the literature on fear memory reconsolidation, there are certain limitations: (i) This study lacks a multi-measure physiological and neurochemical assessment approach, which would allow for a more detailed and comprehensive analysis of the results. (ii) In this study, the music stimuli had positive valence and low arousal. However, examining the effect of music with high arousal and positive valence on disrupting the reconsolidation process and reducing fear-related responses is also essential. (iii) Another limitation of the study is that we cannot rule out the fact that the neural mechanisms for music

memory and extinction memory are non-identical and dissimilar, suggesting the involvement of a distinct neural circuitry during the reconsolidation window. Therefore, future studies may use neuroimaging and electrophysiological techniques to understand the underlying processes of memory reconsolidation with music intervention. (iv) In this study, we did not measure the effect of time on the return of fear, i.e., spontaneous recovery, which needs to be addressed in future research. (v) The effect of music on reducing fear memories may also be studied using a music stimulus according to the participants' musical preference rather than a stimulus selected by the experimenter.

## Conclusion

In conclusion, the study introduces a new approach to modifying fear memories using music during the reconsolidation window. Our findings demonstrated that using music with positive valence and low arousal after reactivation, followed by extinction (during the reconsolidation window), attenuated fear responses during the extinction phase and prevented the return of fear. Here, we propose a novel drug-free intervention to aid in updating the fear memories upon reactivation. Hence, this study contributes to the growing body of research in memory reconsolidation and shows the potential of music in updating fear memories. It further opens avenues for future research to refine and expand upon these findings in a healthy and clinical population.

## Author Contributions

**Conceptualization:** Ankita Verma, Manish Kumar Asthana.

**Data curation:** Ankita Verma.

**Formal analysis:** Ankita Verma, Sharmili Mitra, Manish Kumar Asthana.

**Investigation:** Ankita Verma, Manish Kumar Asthana.

**Methodology:** Ankita Verma, Manish Kumar Asthana.

**Project administration:** Manish Kumar Asthana.

**Resources:** Manish Kumar Asthana.

**Software:** Manish Kumar Asthana.

**Supervision:** Manish Kumar Asthana.

**Validation:** Manish Kumar Asthana.

**Visualization:** Ankita Verma, Sharmili Mitra.

**Writing – original draft:** Ankita Verma, Sharmili Mitra.

**Writing – review & editing:** Abdulrahman Khamaj, Vivek Kant, Manish Kumar Asthana.

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
