## [Decision Letter · Decision Letter 0]

17 Aug 2023

PONE-D-23-20379Music prevents the return of fear in humans using reactivation-extinction paradigmPLOS ONE

Dear Dr. Asthana,

Thank you for submitting your manuscript to PLOS ONE. After careful consideration, we feel that it has merit but does not fully meet PLOS ONE’s publication criteria as it currently stands. Therefore, we invite you to submit a revised version of the manuscript that addresses the points raised during the review process.

We look forward to receiving your revised manuscript.

Kind regards,

Simone Battaglia

Guest Editor

PLOS ONE

Reviewers' comments:

Reviewer's Responses to Questions

**Comments to the Author**

1. Is the manuscript technically sound, and do the data support the conclusions?

Reviewer #1: Yes

Reviewer #2: Yes

2. Has the statistical analysis been performed appropriately and rigorously? 

Reviewer #1: Yes

Reviewer #2: Yes

3. Have the authors made all data underlying the findings in their manuscript fully available?

Reviewer #1: Yes

Reviewer #2: Yes

4. Is the manuscript presented in an intelligible fashion and written in standard English?

Reviewer #1: Yes

Reviewer #2: Yes

5. Review Comments to the Author

Reviewer #1: 7 August 2023

The review on the manuscript, titled “Music prevents the return of fear in humans using reactivation-extinction paradigm” by Verma A et al., submitted to Plos One

Manuscript ID: PONE-D-23-20379

Dear Authors,

Developing effective and accessible interventions for anxiety disorders that can prevent the return of fear is one of the current challenges. In the present research article titled “Music prevents the return of fear in humans using reactivation-extinction paradigm,” Verma and colleagues explore the recovery of this inhibition through the use of the citrus flavonoid hesperetin. The authors explore the use of music to prevent the return of fear in humans by using a reactivation-extinction paradigm and present promising results for the use of music as an intervention technique.

The primary strength of this manuscript is that it provides promising results and contributes to the growing body of research on the use of music in the treatment of anxiety disorders.

In general, I think the idea of this article is really interesting, and the authors’ fascinating observations on this timely topic may be of interest to the readers of Plos One. However, some comments, as well as some crucial evidence that should be included to support the authors’ argumentation, needed to be addressed to improve the quality of the manuscript, its adequacy, and its readability prior to its publication in the present form. My overall opinion is to publish this research article after the authors have carefully considered my comments and suggestions below.

Please consider the following comments:

1. I recommend revising the title. Suggestion: “Music as a Therapeutic Tool: Preventing the Return of Fear in Humans Using Reactivation-Extinction Paradigm " [1 –3].

2. A graphical abstract that will visually summarize the main findings of the manuscript is highly recommended.

3. Abstract: I would like the authors to make as much effort for this section as for the rest of the manuscript. Please present the abstract in 200 words (preferably 200–220 words, max. 250), although the journal allows 300 words [4], focusing on proportionally presenting the background, methods, results, and conclusion (without the headings of subsections). The background should include the general background (one to two sentences), the specific background (two to three sentences), and "the current issue addressed to this study" (one sentence), leading to the objectives. Additionally, the methods should clarify the authors’ approach, such as study design and variables, to solving the problem and/or making progress on the problem. The results should close with a single sentence putting the results in a more general context. The conclusion should open with one sentence describing the main result using such words as “Here we show”, which should be followed by statements such as the potential and the advance this study has provided in the field, and finally a broader perspective (two to three sentences) readily comprehensible to a scientist in any discipline [5–8].

4. Keywords: Please list as many keywords as allowed by the journal, chosen from Medical Subject Headings (MeSH) [9] and use as many as possible in the title and in the first two sentences of the abstract [7,8].

5. Introduction: The authors need to fully expand this section with several paragraphs made up of about 1000 words, introducing information on the main constructs of this protocol, which should be understood by a reader in any discipline, and making it persuasive enough to put forward the main purpose of the current research the authors have conducted and the specific purpose the authors have intended by this study. I would like to encourage the authors to present the introduction starting with the general background, proceeding to the specific background, and finally the current issue addressed to this study, leading to the objectives. Those main structures should be organized in a logical and cohesive manner [10].

6. In this regard, the manuscript would greatly benefit from incorporating a discussion about the underlying neural substrates of memory consolidation and fear memory formation. The following works, but not limited to, may enhance the value of this manuscript [11–15].

7. Methods: I recommend opening this section with a short introductory paragraph and citing more references to ensure the reliability and integrity of the evidence in the study design the authors built and the methodology they have decided to apply.

8. Results: Please refrain from describing statistical values in the body of the text; tables should be used instead. I recommend the authors present figures in color. I suggest closing this section with a paragraph that puts the results into a more general context.

9. Discussion: The discussion section lacks a clear and structured organization. I would like the authors to expand this section into several paragraphs of about 1500 words, beginning this section with an introduction and providing a summary of the previous section. Then, I expect the authors to develop arguments clarifying the potential of this study as an extension of the previous work, the implication of the findings, how this study could facilitate future research, the ultimate goal, the challenge, the knowledge and technology necessary to achieve this goal, the statement about this field in general, and finally the importance of this line of research. It is particularly important to present its limits, its merits, and the potential translation of this study into clinical practice [16,17].

10. Conclusion: I believe that presenting this section with 150–200 words would benefit from a single paragraph that presents some thoughtful and in-depth considerations by the authors as experts in order to convey the main message. The authors should make an effort to explain the theoretical implications as well as the translational application of their research. In order to understand the significance of this field, I believe it would be necessary to discuss theoretical and methodological avenues in need of refinement as well as future research directions.

Overall, the manuscript contains two figures, two tables, and 87 references. I believe that the manuscript has several merits, including its well-designed study that explores the use of music as an intervention technique to prevent the return of fear in humans, using a reactivation-extinction paradigm. The study provides promising results and contributes to the growing body of research on the use of music in the treatment of anxiety disorders. The manuscript is well-organized and provides clear explanations of the study design, methods, and results, making it accessible to a wide range of readers, presenting a valuable resource for anyone interested in the use of music as an intervention technique for anxiety disorders. I hope that, after careful revisions, the manuscript can meet the journal’s high standards for publication.

I declare no conflict of interest regarding this manuscript.

Best regards,

Reviewer

References:

1. https://plos.org/resource/how-to-write-a-great-title/

2. https://www.nature.com/nature-index/news-blog/how-to-write-a-good-research-science-academic-paper-title

3. https://www.indeed.com/career-advice/career-development/catchy-title

4. https://journals.plos.org/plosone/s/submission-guidelines

5. https://www.scribbr.com/dissertation/abstract/

6. https://writing.wisc.edu/handbook/assignments/writing-an-abstract-for-your-research-paper/

7. https://doi.org/10.5812/ijem.100159

8. https://doi.org/10.4103/sja.SJA_685_18

9. https://meshb.nlm.nih.gov/

10. https://dept.writing.wisc.edu/wac/writing-an-introduction-for-a-scientific-paper/

11. https://doi.org/10.1016/j.brs.2023.01.028

12. https://doi.org/10.3389/fnmol.2023.1217090

13. https://doi.org/10.1111/psyp.14122

14. https://doi.org/10.3390/ijms24065926

15. https://doi.org/10.3389/fpsyt.2023.1225755

16. https://doi.org/10.3163/1536-5050.103.2.001

17. https://www.scribbr.com/dissertation/discussion/

Reviewer #2: Asthana and colleagues, in the present article titled ‘Music prevents the return of fear in humans using reactivation-extinction paradigm’ investigated a novel approach to managing fear memories using music of positive valence and low arousal during the reconsolidation window. The study is conducted in the context of fear conditioning, with the aim of finding a drug-free intervention to update fear memories. The research compares the effectiveness of standard extinction (SE), reactivation-extinction (RE), and music reactivation-extinction (MRE) groups in attenuating fear-related responses and preventing the return of fear. The study employs a five-phase experimental design: habituation, acquisition, extinction, reinstatement, and re-extinction of fear. The results indicate that both RE and MRE groups are more effective in attenuating fear-related responses during extinction and re-extinction phases compared to standard extinction. While the study doesn't show a significant difference between SE and RE groups in the return of fear, a trend suggests that RE might be more effective in attenuating return of fear responses. The combination of music and extinction during the reconsolidation window (MRE group) seems particularly effective in preventing the return of fear upon reinstatement. The discussion highlights the potential of the music intervention to disrupt fear-related memories within the reconsolidation window. The study's results suggest that combining music with reactivation and extinction could be more effective than reactivation-extinction alone in reducing fear responses.

In conclusion, the study introduces a new approach to modifying fear memories using music during reconsolidation. The findings suggest that the music intervention could effectively attenuate fear responses and prevent their return. The study provides insights into the potential of music-based interventions in psychological therapies and opens avenues for future research to refine and expand upon these findings. In general, I think the idea of this article is really interesting and the authors’ fascinating observations on this timely topic may be of interest to the readers of Plos One. However, some comments, as well as some crucial evidence that should be included to support the author’s argumentation, needed to be addressed to improve the quality of the manuscript, its adequacy, and its readability prior to the publication in the present form. My overall judgment is to publish this paper after the authors have carefully considered my suggestions below, in particular reshaping parts of the ‘Introduction’ and ‘Methods’ sections by adding more evidence.

Please consider the following comments:

• I recommend revising the title. While it is generally clear, it might benefit from a slight rephrasing to enhance clarity. Consider rewording it to explicitly convey that music is used as an intervention during the reconsolidation window. For example, "Preventing Fear Return in Humans: Music-Based Intervention During Reactivation-Extinction Paradigm." [1-3].

• Abstract: According to the Journal’s guidelines, this section should be presented as a short summary of about 200 words maximum that objectively represents the article. It should let readers get the gist or essence of the manuscript quickly, prepare the readers to follow the detailed information, analyses, and arguments in the full paper and, most of all, it should help readers remember key points from your paper. Please, consider rewrite this paragraph following these instructions [4].

• Keywords: Please list ten keywords chosen from Medical Subject Headings (MeSH) and use as many as possible in the title and in the first two sentences of the abstract. I would suggest adding “Neuroplasticity” and “Emotion regulation” as keywords.

• Introduction: The authors need to reorganize this section with several paragraphs made up of about 1000 words, introducing information on the main constructs of this study, which should be understood by a reader in any discipline, and making it persuasive enough to put forward the main purpose of the current research the authors have conducted and the specific purpose the authors have intended by this protocol. I would like to encourage the authors to present the introduction starting with the general background, proceeding to the specific background on fear-related memories, fear extinction, memory reconsolidation, and the potential of music as a non-pharmacological intervention to prevent the return of fear responses. Those main structures should be organized in a logical and cohesive manner [5].

• In this regard, I believe that the Introduction section would benefit from additional information to enhance its clarity and contextualization. Therefore, to enhance the depth of understanding, I would recommend incorporating a brief discussion on the neural substrates underlying fear memory formation, consolidation, and modulation. Understanding the neurobiological basis of fear-related memories is crucial in contextualizing the proposed intervention involving music and memory reconsolidation. Research has shown that fear memories involve intricate neural networks, primarily centered around the amygdala, a key hub for processing emotional stimuli and generating fear responses. The amygdala's lateral nucleus (LA) is particularly crucial in associating the conditioned stimulus (CS) with the unconditioned stimulus (UCS), forming the basis of fear memory formation. Furthermore, the central nucleus of the amygdala (CeA) orchestrates the expression of fear responses through its connections with brainstem regions responsible for autonomic and behavioral responses to threats [6-7]. Consolidation of fear memories involves complex interactions between the amygdala and the hippocampus. The hippocampus encodes contextual information and contributes to the formation of episodic memories related to fear-inducing events. The prefrontal cortex, including the medial prefrontal cortex (mPFC), plays a role in regulating fear responses through its inhibitory control over the amygdala. This neural circuitry's dynamic interplay contributes to fear memory consolidation, retention, and subsequent retrieval. Regarding modulation, memory reconsolidation theory suggests that reactivation of a fear memory engages the same neural circuits that were involved in its initial formation. During reconsolidation, synaptic plasticity mechanisms and protein synthesis contribute to the updating and modification of the memory trace. This offers a potential avenue for interventions like the one proposed in this study, involving the exposure to music during the reconsolidation window. Incorporating a discussion of these neural substrates provides a neuroscientific foundation for understanding how interventions like music might interact with fear memory processes [8-10]. It highlights the intricate interplay between brain regions that could be influenced by emotional stimuli such as music, shedding light on the potential mechanisms underlying the observed effects. Such an addition would enhance the paper's interdisciplinary perspective and contribute to a holistic understanding of the study's implications.

• Material and Methods: I believe that this section would benefit from a clearer structure and better organization of the flow of information. For example, I suggest adding a brief explanation of why Indian Institute of Technology Roorkee was chosen as the recruitment site. This can help readers understand the context. Also, while the power analysis is mentioned, it would be helpful to explain why the chosen effect size of 0.25 was considered appropriate for this study.

• Discussion: While the interpretation of findings is generally clear, it's crucial to discuss both the practical and theoretical implications of the observed effects. How might the results contribute to our understanding of memory reconsolidation and its potential applications? Furthermore, when discussing the effectiveness of the music intervention, authors might provide a more detailed comparison with similar studies using different interventions post-reactivation that would help readers understand the unique contribution of your approach.

• In my opinion, the ‘Conclusions’ paragraph would benefit from some thoughtful as well as in-depth considerations by the authors, because as it stands, it lists down all the main findings of the research, without really stressing the theoretical significance of the study. Authors should make an effort, trying to explain the theoretical implication as well as the translational application of their research.

• Please consider providing more specific suggestions for future research directions. For instance, how might the study be expanded to address the limitations mentioned, and what potential hypotheses or questions could arise from the current findings?

• Finally, while the language is generally clear, strive for a more concise and focused style. Avoid overly lengthy sentences that may obscure the key points.

I hope that, after these careful revisions, the manuscript can meet the Journal’s high standards for publication. I am available for a new round of revision of this article.

I declare no conflict of interest regarding this manuscript.

Best regards,

Reviewer

References:

1. https://plos.org/resource/how-to-write-a-great-title/

2. https://www.nature.com/nature-index/news-blog/how-to-write-a-good-research-science-academic-paper-title

3. https://www.indeed.com/career-advice/career-development/catchy-title

4. https://journals.plos.org/plosone/s/submission-guidelines

5. https://dept.writing.wisc.edu/wac/writing-an-introduction-for-a-scientific-paper/

6. DOI: 10.17219/acem/165944

7. https://doi.org/10.3390/ijms24065926

8. DOI: 10.3390/biomedicines11030945

9. https://doi.org/10.3389/fnmol.2023.1217090

10. https://doi.org/10.3390/biomedicines11051248

6. PLOS authors have the option to publish the peer review history of their article (what does this mean?). If published, this will include your full peer review and any attached files.

Reviewer #1: No

Reviewer #2: No

---

## [Author Response · Author response to Decision Letter 0]

30 Sep 2023

Response to Reviewers

As per the reviewers’ comments, we have incorporated the changes in our manuscript. We have highlighted the changes in yellow in the ‘Revised Manuscript with Track Changes’ document. Please find the responses to the reviewers’ comments as follows:

Response to Reviewer 1

 1. I recommend revising the title. Suggestion: “Music as a Therapeutic Tool: Preventing the Return of Fear in Humans Using Reactivation-Extinction Paradigm " [1 –3].

Response- I appreciate your suggestion to reconsider the title and would like to thank you for your feedback. With your and another reviewer's input, we have decided to rename the paper "Preventing Fear Return in Humans: Music-Based Intervention During Reactivation-Extinction Paradigm"

 2. A graphical abstract that will visually summarize the main findings of the manuscript is highly recommended.

Response- I sincerely value your suggestion and acknowledge the importance of incorporating a graphical abstract to effectively communicate the main findings of the manuscript. However, I have been unable to locate any existing research papers that could serve as a reference for guidance in this matter. If feasible, I kindly request that you provide a reference paper, as it would greatly assist in addressing this concern.

3. Abstract: I would like the authors to make as much effort for this section as for the rest of the manuscript. Please present the abstract in 200 words (preferably 200–220 words, max. 250), although the journal allows 300 words [4], focusing on proportionally presenting the background, methods, results, and conclusion (without the headings of subsections). The background should include the general background (one to two sentences), the specific background (two to three sentences), and "the current issue addressed to this study" (one sentence), leading to the objectives. Additionally, the methods should clarify the authors’ approach, such as study design and variables, to solving the problem and/or making progress on the problem. The results should close with a single sentence putting the results in a more general context. The conclusion should open with one sentence describing the main result using such words as “Here we show”, which should be followed by statements such as the potential and the advance this study has provided in the field, and finally a broader perspective (two to three sentences) readily comprehensible to a scientist in any discipline [5–8].

Response- Dear reviewer, Thank you for your feedback. We have made efforts to adhere to the prescribed word limit of 220 words for the abstract. As per your recommendation, we have eliminated all subsection headings. All of the perceptive ideas regarding the structuring of the abstract have been implemented to the fullest extent of our understanding.

 4. Keywords: Please list as many keywords as allowed by the journal, chosen from Medical Subject Headings (MeSH) [9] and use as many as possible in the title and in the first two sentences of the abstract [7,8].

Response- Dear reviewer, based on your insightful comments we have used ten different keywords based on the suggestion from Medical Subject Headings and our own understanding of the construct. 

 5. Introduction: The authors need to fully expand this section with several paragraphs made up of about 1000 words, introducing information on the main constructs of this protocol, which should be understood by a reader in any discipline, and making it persuasive enough to put forward the main purpose of the current research the authors have conducted and the specific purpose the authors have intended by this study. I would like to encourage the authors to present the introduction starting with the general background, proceeding to the specific background, and finally the current issue addressed to this study, leading to the objectives. Those main structures should be organized in a logical and cohesive manner [10].

Response- Dear reviewer, we would like to extend our appreciation for the valuable input that has been provided. Based on the valuable feedback and insightful observations provided, we have expanded the introductory paragraph to encompass a total of 1302 words (Page no. 5, lines 20-25, Page no. 6, lines 1-13).This research includes an additional section that elaborates upon the involvement of several amygdala nuclei in relation to fear conditioning. Moreover, we delved into further physiological intricacies about the pivotal involvement of the hippocampus in the process of fear-related memory formation. The expanded section is highlighted in yellow in the ‘Revised Manuscript with Track Changes’.

 6. In this regard, the manuscript would greatly benefit from incorporating a discussion about the underlying neural substrates of memory consolidation and fear memory formation. The following works, but not limited to, may enhance the value of this manuscript [11–15].

Response- Dear reviewer, we would like to extend our appreciation for the valuable input that has been provided. Based on the valuable feedback and insightful observations provided, we have expanded the introductory paragraph to encompass a total of 1302 words (Page no. 5, lines 20-25, Page no. 6, lines 1-13).This research includes an additional section that elaborates upon the involvement of several amygdala nuclei in relation to fear conditioning. Moreover, we delved into further physiological intricacies about the pivotal involvement of the hippocampus in the process of fear-related memory formation. The expanded section is highlighted in yellow in the ‘Revised Manuscript with Track Changes’.

 7. Methods: I recommend opening this section with a short introductory paragraph and citing more references to ensure the reliability and integrity of the evidence in the study design the authors built and the methodology they have decided to apply.

Response- Dear reviewer, in accordance with the suggestions, we have implemented the aforementioned changes in our present manuscript. Specifically, we have included references to the works of Schiller et al. (2012), Fernandez-Rey et al. (2018), and Ganho-Ávila et al. (2019). These references have been explicitly cited and emphasized on page number 10, line numbers 10 and 11.

 8. Results: Please refrain from describing statistical values in the body of the text; tables should be used instead. I recommend the authors present figures in color. I suggest closing this section with a paragraph that puts the results into a more general context.

Response- Dear reviewer on the basis of your suggestions, we have included the color figures in our manuscript. However, while the suggestion to use statistical values in a table is a good one, I lament to inform you that it may compromise the structure of my paper. If you are not comfortable with this, please let us know and we will revise it further. 

 9. Discussion: The discussion section lacks a clear and structured organization. I would like the authors to expand this section into several paragraphs of about 1500 words, beginning this section with an introduction and providing a summary of the previous section. Then, I expect the authors to develop arguments clarifying the potential of this study as an extension of the previous work, the implication of the findings, how this study could facilitate future research, the ultimate goal, the challenge, the knowledge and technology necessary to achieve this goal, the statement about this field in general, and finally the importance of this line of research. It is particularly important to present its limits, its merits, and the potential translation of this study into clinical practice [16,17].

Response- Dear reviewer, In response to your insightful ideas, we have made significant revisions to the discussion part, resulting in an expansion to a total of 1505 words. We have taken into careful consideration all of the great comments provided and have integrated them accordingly within this section. Attention was paid in adding more relevant and pertinent literature on music intervention, a section on the important brain structure that was involved in memory consolidation and reconsolidation process was also incorporated. In addition, the authors made an effort to elaborate on the merits and limits of the current study, as well as the application of the findings in clinical settings and in normal populations. The changes are highlighted in yellow on Page no 16, lines 5-8; page no. 18, lines 21-25; page no 19, lines 1-3; page no 19, lines 9-21; page no 20, lines 21-24 of the ‘Revised Manuscript with Track Changes’ document.

 10. Conclusion: I believe that presenting this section with 150–200 words would benefit from a single paragraph that presents some thoughtful and in-depth considerations by the authors as experts in order to convey the main message. The authors should make an effort to explain the theoretical implications as well as the translational application of their research. In order to understand the significance of this field, I believe it would be necessary to discuss theoretical and methodological avenues in need of refinement as well as future research directions.

Response- Dear reviewer, in response to your recommendation, we have incorporated a dedicated section for the conclusion in this manuscript. In this part, we made an attempt to explain how the findings of this research may be applied to settings other than those used for experiments, as well as its practical applications and the likelihood that its findings can be replicated on clinical and non-clinical populations. The changes are highlighted on page no 21, line no 3-12 of the ‘Revised Manuscript with Track Changes’ document.

Response to Reviewer 2

1. I recommend revising the title. While it is generally clear, it might benefit from a slight rephrasing to enhance clarity. Consider rewording it to explicitly convey that music is used as an intervention during the reconsolidation window. For example, "Preventing Fear Return in Humans: Music-Based Intervention During Reactivation-Extinction Paradigm." [1-3].

Response- I appreciate your suggestion to reconsider the title and would like to thank you for your feedback. With your and another reviewer's input, we have decided to rename the paper "Preventing Fear Return in Humans: Music-Based Intervention During Reactivation-Extinction Paradigm"

 2. Abstract: According to the Journal’s guidelines, this section should be presented as a short summary of about 200 words maximum that objectively represents the article. It should let readers get the gist or essence of the manuscript quickly, prepare the readers to follow the detailed information, analyses, and arguments in the full paper and, most of all, it should help readers remember key points from your paper. Please, consider rewriting this paragraph following these instructions [4].

Response- Dear reviewer, Thank you for your feedback. We have made efforts to make the abstract as concise as possible. We have been able to present the summary of the article in 220 words, adhering to the prescribed word limit of 200-300 words for the abstract. If further modifications are required, please let us know and we will revise it accordingly. 

 3. Keywords: Please list ten keywords chosen from Medical Subject Headings (MeSH) and use as many as possible in the title and in the first two sentences of the abstract. I would suggest adding “Neuroplasticity” and “Emotion regulation” as keywords.

Response- Dear reviewer, based on your insightful comments we have used ten different keywords based on the suggestion from Medical Subject Headings and our own understanding of the construct. 

 4. Introduction: The authors need to reorganize this section with several paragraphs made up of about 1000 words, introducing information on the main constructs of this study, which should be understood by a reader in any discipline, and making it persuasive enough to put forward the main purpose of the current research the authors have conducted and the specific purpose the authors have intended by this protocol. I would like to encourage the authors to present the introduction starting with the general background, proceeding to the specific background on fear-related memories, fear extinction, memory reconsolidation, and the potential of music as a non-pharmacological intervention to prevent the return of fear responses. Those main structures should be organized in a logical and cohesive manner [5].

 In this regard, I believe that the Introduction section would benefit from additional information to enhance its clarity and contextualization. Therefore, to enhance the depth of understanding, I would recommend incorporating a brief discussion on the neural substrates underlying fear memory formation, consolidation, and modulation. Understanding the neurobiological basis of fear-related memories is crucial in contextualizing the proposed intervention involving music and memory reconsolidation. Research has shown that fear memories involve intricate neural networks, primarily centered around the amygdala, a key hub for processing emotional stimuli and generating fear responses. The amygdala's lateral nucleus (LA) is particularly crucial in associating the conditioned stimulus (CS) with the unconditioned stimulus (UCS), forming the basis of fear memory formation. Furthermore, the central nucleus of the amygdala (CeA) orchestrates the expression of fear responses through its connections with brainstem regions responsible for autonomic and behavioral responses to threats [6-7]. Consolidation of fear memories involves complex interactions between the amygdala and the hippocampus. The hippocampus encodes contextual information and contributes to the formation of episodic memories related to fear-inducing events. The prefrontal cortex, including the medial prefrontal cortex (mPFC), plays a role in regulating fear responses through its inhibitory control over the amygdala. This neural circuitry's dynamic interplay contributes to fear memory consolidation, retention, and subsequent retrieval. Regarding modulation, memory reconsolidation theory suggests that reactivation of a fear memory engages the same neural circuits that were involved in its initial formation. During reconsolidation, synaptic plasticity mechanisms and protein synthesis contribute to the updating and modification of the memory trace. This offers a potential avenue for interventions like the one proposed in this study, involving the exposure to music during the reconsolidation window. Incorporating a discussion of these neural substrates provides a neuroscientific foundation for understanding how interventions like music might interact with fear memory processes [8-10]. It highlights the intricate interplay between brain regions that could be influenced by emotional stimuli such as music, shedding light on the potential mechanisms underlying the observed effects. Such an addition would enhance the paper's interdisciplinary perspective and contribute to a holistic understanding of the study's implications.

Response- Thank you for your valuable insights on this section. As per your comments, we have incorporated the section on neural substrates suggested by you, and have also included some references for the same. We are extremely grateful for your valued additions to this section of our manuscript. The changes are highlighted on Page no. 5, lines 20-25, Page no. 6, lines 1-13 of the ‘Revised Manuscript with Track Changes’ document.

 5. Material and Methods: I believe that this section would benefit from a clearer structure and better organization of the flow of information. For example, I suggest adding a brief explanation of why Indian Institute of Technology Roorkee was chosen as the recruitment site. This can help readers understand the context. Also, while the power analysis is mentioned, it would be helpful to explain why the chosen effect size of 0.25 was considered appropriate for this study.

Response- We thank you for your valuable comment. We have chosen Indian Institute of Technology as the recruitment site for the ease of data collection. Since our study comprises a 3-day paradigm, this sampling method was chosen for its convenience and to minimize the likelihood of participant attrition. However, since the site of data collection is a national institute, our participants belong to diverse socio-cultural backgrounds; Hence, the generalizability of the study is not compromised.

Further, we have chosen a medium effect size of 0.25 based on previous studies on fear conditioning and we have provided a reference for our sample size selection in the methods section, highlighted on page no 7, lines 7-11 of the ‘Revised Manuscript with Track Changes’ document.

 6. Discussion: While the interpretation of findings is generally clear, it's crucial to discuss both the practical and theoretical implications of the observed effects. How might the results contribute to our understanding of memory reconsolidation and its potential applications? Furthermore, when discussing the effectiveness of the music intervention, authors might provide a more detailed comparison with similar studies using different interventions post-reactivation that would help readers understand the unique contribution of your approach.

Response- Dear reviewer, In response to your insightful ideas, we have made significant revisions to the discussion part, resulting in an expansion to a total of 1505 words. We have taken into careful consideration all of the great comments provided and have integrated them accordingly within this section. Attention was paid in adding more relevant and pertinent literature on music intervention, a section on the important brain structure that was involved in memory consolidation and reconsolidation process was also incorporated. In addition, the authors made an effort to elaborate on the merits and limits of the current study, as well as the application of the findings in clinical settings and in normal populations. The changes are highlighted in yellow on Page no 16, lines 5-8; page no. 18, lines 21-25; page no 19, lines 1-3; page no 19, lines 9-21; page no 20, lines 21-24 of the ‘Revised Manuscript with Track Changes’ document.

 7. In my opinion, the ‘Conclusions’ paragraph would benefit from some thoughtful as well as in-depth considerations by the authors, because as it stands, it lists down all the main findings of the research, without really stressing the theoretical significance of the study. Authors should make an effort, trying to explain the theoretical implication as well as the translational application of their research.

Response- Dear reviewer, in response to your recommendation, we have incorporated a dedicated section for the conclusion in this manuscript. In this part, we made an attempt to explain how the findings of this research may be applied to settings other than those used for experiments, as well as its practical applications and the likelihood that its findings can be replicated on clinical and non-clinical populations. The changes are highlighted on page no 21, line no 3-12 of the ‘Revised Manuscript with Track Changes’ document.

 8. Please consider providing more specific suggestions for future research directions. For instance, how might the study be expanded to address the limitations mentioned, and what potential hypotheses or questions could arise from the current findings? Finally, while the language is generally clear, strive for a more concise and focused style. Avoid overly lengthy sentences that may obscure the key points.

Response- We thank you for your valuable feedback. We have added the limitations and future directions, highlighted on page no. 20, lines 21-24.

---

## [Decision Letter · Decision Letter 1]

23 Oct 2023

Preventing Fear Return in Humans: Music-Based Intervention During Reactivation-Extinction Paradigm

PONE-D-23-20379R1

Dear Dr. Asthana,

We’re pleased to inform you that your manuscript has been judged scientifically suitable for publication and will be formally accepted for publication once it meets all outstanding technical requirements.

Kind regards,

Simone Battaglia

Guest Editor

PLOS ONE

Additional Editor Comments (optional):

Reviewers' comments:

Reviewer's Responses to Questions

**Comments to the Author**

1. If the authors have adequately addressed your comments raised in a previous round of review and you feel that this manuscript is now acceptable for publication, you may indicate that here to bypass the “Comments to the Author” section, enter your conflict of interest statement in the “Confidential to Editor” section, and submit your "Accept" recommendation.

Reviewer #1: All comments have been addressed

Reviewer #2: All comments have been addressed

2. Is the manuscript technically sound, and do the data support the conclusions?

Reviewer #1: Yes

Reviewer #2: Yes

3. Has the statistical analysis been performed appropriately and rigorously? 

Reviewer #1: Yes

Reviewer #2: Yes

4. Have the authors made all data underlying the findings in their manuscript fully available?

Reviewer #1: Yes

Reviewer #2: Yes

5. Is the manuscript presented in an intelligible fashion and written in standard English?

Reviewer #1: Yes

Reviewer #2: Yes

6. Review Comments to the Author

Reviewer #1: 20 October 2023

The 2nd review on the manuscript, titled “Music prevents the return of fear in humans using reactivation-extinction paradigm” by Verma A et al., submitted to Plos One

Manuscript ID: PONE-D-23-20379

Dear Authors,

I am pleased to see that the authors have addressed the issues I raised in the previous round. Currently, the manuscript is a well-written research paper with informative layouts that presents promising results and contributes to the growing body of research on the use of music in the treatment of anxiety disorders. I believe the manuscript meets the journal’s high standards for publication. I am looking forward to seeing more papers written by the same authors.

Thank you, I declare no conflict of interest regarding this manuscript.

Best regards,

Reviewer

Reviewer #2: Dear Authors,

Thank you for your thoughtful responses to the comments I provided on the revised manuscript.

After careful consideration, I am satisfied with the revisions you have made and your responses to my concerns. Therefore, I am confirming my acceptance of the manuscript for publication. I believe your efforts have improved the quality of the paper, and I look forward to seeing it contribute to the scientific community.

Best regards,

Reviewer

7. PLOS authors have the option to publish the peer review history of their article (what does this mean?). If published, this will include your full peer review and any attached files.

Reviewer #1: No

Reviewer #2: No

---

## [Editor Report · Acceptance letter]

28 Jan 2024

PONE-D-23-20379R1 

PLOS ONE

Dear Dr. Asthana, 

I'm pleased to inform you that your manuscript has been deemed suitable for publication in PLOS ONE. Congratulations! Your manuscript is now being handed over to our production team.

Kind regards, 

on behalf of

Dr. Simone Battaglia 

Guest Editor

PLOS ONE